# Prediction of Failure Modes and Minimum Characteristic Value of Transverse Reinforcement of RC Beams Based on Interpretable Machine Learning

Sixuan Wang [1,2], Cailong Ma [1,2,3,*], Wenhu Wang [1,3], Xianlong Hou [1,3], Xufeng Xiao [2], Zhenhao Zhang [4], Xuanchi Liu [5] and JinJing Liao [6,*]

1   School of Civil Engineering and Architecture, Xinjiang University, Urumqi 830047, China
2   College of Mathematics and System Sciences, Xinjiang University, Urumqi 830047, China
3   Xinjiang Key Lab of Building Structure and Earthquake Resistance, Xinjiang University, Urumqi 830047, China
4   School of Civil Engineering, Changsha University of Science and Technology, Changsha 410114, China
5   Department of Infrastructure Engineering, The University of Melbourne, Parkville, VIC 3010, Australia
6   School of Civil and Transportation Engineering, Guangdong University of Technology, Guangzhou 510006, China
*   Correspondence: macailong@xju.edu.cn (C.M.); liaojj@gdut.edu.cn (J.L.); Tel.: +86-991-858-2256 (C.M.)

**Abstract:** Shear failure of reinforced concrete (RC) beams is a form of brittle failure and has always been a concern. This study adopted the interpretable machine-learning technique to predict failure modes and identify the boundary value between different failure modes to avoid diagonal splitting failure. An experimental database consisting of 295 RC beams with or without transverse reinforcements was established. Two features were constructed to reflect the design characteristics of RC beams, namely, the shear–span ratio and the characteristic value of transverse reinforcement. The characteristic value of transverse reinforcement has two forms: (i) $\lambda_{sv,f_t} = \rho_{stp} f_{sv} / f_t$, from the China design code of GB 50010-2010; and (ii) $\lambda_{sv,f_c'} = \rho_{stp} f_{sv} / f_c'^{0.5}$, from the America design code of ACI 318-19 and Canada design code of CSA A23.3-14. Six machine-learning models were developed to predict failure modes, and gradient boosting decision tree and extreme gradient boosting are recommended after comparing the prediction performance. Then, shapley additive explanations (SHAP) indicates that the characteristic value of transverse reinforcement has the most significant effect on failure mode, follow by the shear–span ratio. The characteristic value of transverse reinforcement is selected as the form of boundary value. On this basis, an accumulated local effects (ALE) plot describes how this feature affects model prediction and gives the boundary value through numerical simulation, that is, the minimum characteristic value of transverse reinforcement. Compared with the three codes, the suggested value for $\lambda_{sv,f_c',min}$ has higher reliability and security for avoiding diagonal splitting failure. Accordingly, the research approach in this case is feasible and effective, and can be recommended to solve similar tasks.

**Keywords:** reinforced concrete beam; interpretable machine-learning technique; failure modes; the minimum characteristic value of transverse reinforcement; accumulated local effects

## 1. Introduction

Reinforced concrete (RC) structures are the most common and widely used structures. The RC beam is the horizontal member. Under loading, the RC beam generally fails in two ways: normal section failure and inclined section failure [1]. An RC beam with insufficient reinforcement or a small shear–span ratio is prone to shear failure, which is a form of brittle failure. Typical inclined section failure modes of RC beams include diagonal splitting failure, shear compression failure, and diagonal compression failure. Inclined section failure is affected by many factors, including section size, concrete strength, and transverse

reinforcement, in which the shear–span ratio and consumption of transverse reinforcement have the greatest impact [1–9]. An RC beam with a larger shear–span ratio ($a/d > 3$) and lesser transverse reinforcement usually experiences diagonal splitting failure. An RC beam with a smaller shear–span ratio ($a/d < 1$) or moderate shear–span ratio but too much transverse reinforcement usually experiences diagonal compression failure. Shear compression failure falls somewhere in between and is more ductile than the other two. Due to higher brittleness and a faster destruction process, diagonal splitting failure and diagonal compression failure should be avoided in practical engineering. Hence, it is crucial to accurately predict failure modes and identify the boundary between them.

Traditional methods for predicting failure modes are based largely upon experimental failure phenomena at the macroscale. In the judgement of failure modes, there are obvious differences, and no unified recognition or consideration exists. With the accumulation of massive amounts of experimental data and rapid development of artificial intelligence, machine-learning methods are advancing in structural engineering. Especially in recent years, extensive research has been conducted on the failure mode prediction of RC members using machine-learning and data-driven methods [10–17], in which the large amount of experimental data plays a valuable role. These existing research conclusions demonstrate that machine learning can improve inaccurate predictions of traditional methods [10–17], as well as provide a solid research background and basis for this study. Nevertheless, previous studies lack research on model interpretability, which directly impacts the application of machine learning and the validity of prediction results. Explainable artificial intelligence (XAI) is one of the most interesting and challenging research topics in the field of artificial intelligence, particularly when machine learning is applied in biomedical engineering, structural engineering, aircraft structures, etc. The development of XAI is still in its infancy, but it is progressing rapidly, and several tools have been proposed to explain the "black box", such as the partial dependence plot [18], individual conditional expectation plot [19], accumulated local effects plot [20], local interpretable model-agnostic explanations [21], and shapley additive explanations [22]. Recent studies have shown the desirability of XAI in the shear behavior of RC members in terms of several aspects: transparency, informativity, and acceptable model output [14,17,23–33], but do not provide constructive advice combined with the shear mechanism.

To avoid diagonal splitting failure and diagonal compression failure in practical engineering, current codes take the form of controlling the minimum reinforcement ratio and the minimum sectional size, respectively. There are currently regulations regarding the minimum reinforcement ratio, such as $\rho_{stp,min} = 0.240 f_t / f_{yv}$ in the China design code of GB 50010-2010 [34], $\rho_{stp,min} = 0.062 f_c'^{0.5} / f_{yv}$ in the America design code of ACI 318-19 [35], and $\rho_{stp,min} = 0.066 f_c'^{0.5} / f_{yv}$ in the Canada design code of CSA A23.3-14 [36]. These regulations adopt different standards for measuring concrete strength, which can cause errors in judging failure modes. In addition, there are some limitations, such as lower security and reliability. Similar to failure mode prediction, a data-driven approach can also be utilized to identify the boundary value between different failure modes. To date, the researchers have not yet undertaken studies related to this topic, but there are similar studies showing that an accumulated local effects plot can provide data-driven threshold identification. In the bioinformatics field, Lung Yun Teng et al. used shapley additive explanations and an accumulated local effects plot to investigate the effect of the nuchal fold on fetal growth restriction (FGR) and determine its threshold for distinguishing between healthy fetuses and those having FGR [33]. Consequently, the accumulated local effects plot is an effective tool for identifying or even determining a cutoff value [33]. In the structural engineering field, Sujith Mangalathu et al. used a partial dependence plot, accumulated local effects plot, and shapley additive explanations to explore the influence of variables on model prediction and their thresholds in the case of shear strength of shear walls and regional level damage of skewed bridges [31]. Inspired by this, this study attempted to identify the boundary value between different failure modes using a data-driven approach (accumulated local effects plot).

This study aimed to predict the failure modes of RC beams and identify the boundary value between diagonal splitting failure and shear compression failure based on interpretable machine-learning approaches. Through feasibility analysis, novel research ideas and solutions were sought to solve similar issues. Six machine-learning models were applied to predict the failure modes of RC beams, such as random forests, gradient boosting decision tree, and extreme gradient boosting. Then, shapley additive explanations was utilized to assess the importance of features and their influence on model response. On this basis, an accumulated local effects plot was adopted to perform numerical simulation and identify the boundary value for the key feature to prevent diagonal splitting failure.

## 2. Research Thought and Experimental Dataset of RC Beams

### 2.1. Research Thought and Dataset Description

This study mainly focused on the prediction of diagonal splitting failure and shear compression failure, and identification of the threshold between the two, namely, the minimum consumption of transverse reinforcement. The research process diagram is shown in Figure 1. The first step was to establish an experimental dataset of RC beams with or without transverse reinforcements. The second step was to select input features based on the shear mechanism and construct the form of the minimum consumption of transverse reinforcement. The third step was to predict the failure modes of RC beams through training and testing multiple machine-learning models. Finally, the interpretability of the better prediction models was further analyzed, and the minimum consumption of transverse reinforcement was identified and further compared with the safety of traditional regulations.

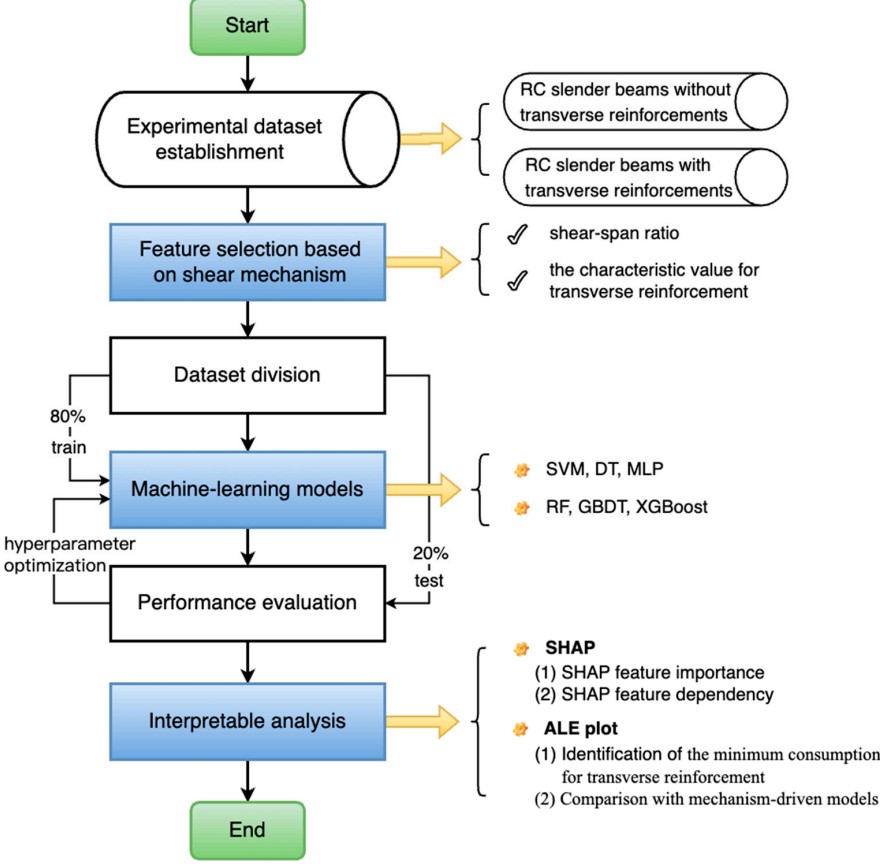

**Figure 1.** The research process diagram of this paper.

In order to carry out the research, an experimental dataset of 295 RC beams with or without transverse reinforcements was established. These specimens were derived from

31 published references (see Table A1 in Appendix A) [37–67]. There were two failure modes in this dataset, namely, diagonal splitting failure (DSF) and shear compression failure (SCF). As shown in Figure 2a, DSF occurred in 87 specimens, accounting for 29.49%, and SCF occurred in 208 specimens, accounting for 70.51%. For 163 RC beams without transverse reinforcements, SCF occurred in 91 specimens, accounting for 55.83%, as shown in Figure 2b. For 132 RC beams with transverse reinforcements, SCF occurred in 117 specimens, accounting for 88.64%, as shown in Figure 2c.

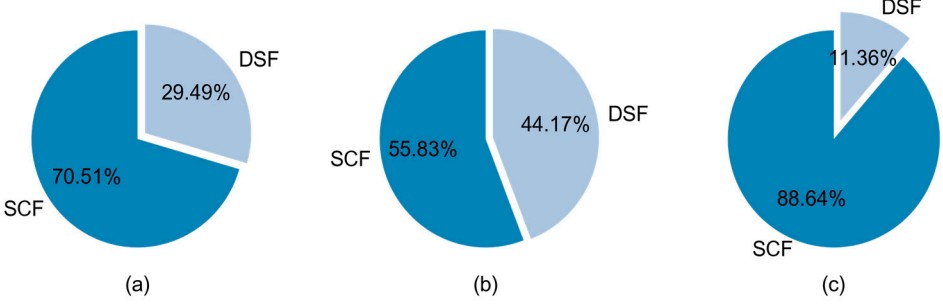

**Figure 2.** Distribution of failure modes in this dataset. (**a**) All RC slender beams; (**b**) RC slender beams without transverse reinforcements; (**c**) RC slender beams with transverse reinforcements.

### 2.2. Feature Selection

Feature engineering is the critical work of developing machine-learning models, which determines the prediction performance to a certain extent. On the basis of shear mechanism codes and mechanism-driven models, the factors affecting the failure mode of RC beams can be summarized into three levels, namely, sectional size, concrete strength grade, and transverse reinforcement consumption [1–9]. The level for the sectional size mainly includes span ($a$) and depth ($d$). The level for the concrete strength grade mainly includes tensile concrete strength ($f_t$) or concrete compression strength ($f_c'$), and the level for transverse reinforcement consumption mainly includes the yield strength for transverse reinforcement ($f_{sv}$) and transverse reinforcement ratio ($\rho_{stp}$). Interactions exist among several influencing factors, so feature selection should consider not only the influencing factors, but also the interactions. Therefore, combined with the above key factors, the input features of machine-learning models adopted in this study are as follows. Table 1 summarizes the statistical information of inputs and Figure 3 shows the distribution of inputs.

**Table 1.** Statistical description of the input features in this dataset.

| Input | Min. | Max. | Mid. | Mean | Std. | Unit |
|---|---|---|---|---|---|---|
| $a/d$ | 2.50 | 5.00 | 3.00 | 3.16 | 0.62 | - |
| $\lambda_{sv,f_t}$ | 0.00 | 1.27 | 0.00 | 0.14 | 0.24 | - |
| $\lambda_{sv,f_c'}$ | 0.00 | 0.73 | 0.00 | 0.07 | 0.12 | - |

Notes: Min. = minimum, Max. = maximum, Mid. = middle, Std. = standard deviance.

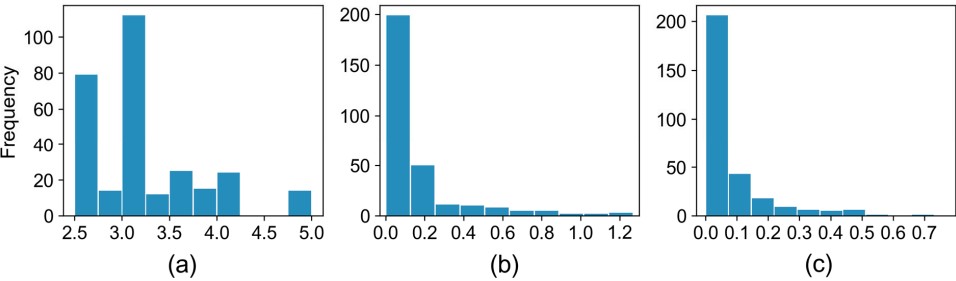

**Figure 3.** Distribution of input features for RC beams in this dataset. (**a**) $\lambda = a/d$; (**b**) $\lambda_{sv,f_t} = \rho_{stp}f_{sv}/f_t$; (**c**) $\lambda_{sv,f_c'} = \rho_{stp}f_{sv}/f_c'$.

- *Shear–span ratio*

The shear–span ratio ($\lambda = a/d$) is a critical factor in the shear strength and failure mode of RC members [1–9,34–36]. The experiments show that when $\lambda > 3$, RC beams without transverse reinforcements are more likely to experience diagonal splitting failure. Therefore, $\lambda$ was selected as the first input feature. As shown in Table 1 and Figure 3, the value of the shear–span ratio ranged from 2.50 to 5.00 and was concentrated in 2.50~2.75 and 3.00~3.25.

- *The characteristic value of transverse reinforcement*

In the shear resistance of RC beams, transverse reinforcement is an essential component. With a transverse reinforcement increase, shear resistance can be enhanced. Furthermore, GB 50010-2010 [34], ACI 318-19 [35], and CSA A23.3-14 [36] also consider the effects of transverse reinforcement on the shear strength and failure mode of RC beams.

In the form of the minimum consumption of transverse reinforcement adopted by GB 50010-2010 [34], this study used the characteristic value of transverse reinforcement based on $f_t$ ($\lambda_{sv,f_t} = \rho_{stp} f_{sv} / f_t$) as the second input feature. As shown in Table 1 and Figure 3, the range of the characteristic value of transverse reinforcement was from 0.00 to 1.27 and most values were less than 0.15.

In the form of the minimum consumption of transverse reinforcement adopted by ACI 318-19 [35] and CSA A23.3-14 [36], the characteristic value of transverse reinforcement based on $f_c'$ ($\lambda_{sv,f_c'} = \rho_{stp} f_{sv} / f_c'^{0.5}$) was used as the third input feature in this study. As shown in Table 1 and Figure 3, it ranged from 0.00 to 0.73 and most values were less than 0.10. Therefore, the combinations were selected as the input features of machine-learning models, as follows:

$$\text{model } 1 \left( \lambda, \ \lambda_{sv,f_t} \right), \tag{1}$$

$$\text{model } 2 \left( \lambda, \ \lambda_{sv,f_c'} \right). \tag{2}$$

## 3. Overview of Machine-Learning Approaches

### 3.1. The Selected Machine-Learning Algorithms

With the rapid development of artificial intelligence, machine learning has been widely used in tabular data and a variety of algorithms have been proposed. Moreover, some studies have used different algorithms to predict failure modes of RC members, in which results show that machine-learning models outperform existing traditional mechanism models in terms of prediction accuracy and generalization ability [10–17].

The primary purpose of this study was to construct an interpretable model for predicting failure modes of RC beams, which is a classification task in supervised learning. In this study, six representative classification algorithms were selected and developed for comparison and evaluation, so as to obtain a model with better performance for further research on the stirrup consumption. The six models selected for this study were: (i) support vector machine (SVM) [68], (ii) decision tree (DT) [68], (iii) multilayer perceptron (MLP) [68,69], (iv) random forests (RF) [70], (v) gradient boosting decision tree (GBDT) [71], and (vi) extreme gradient boosting (XGBoost) [72]. The RF algorithm is a form of ensemble learning based on bagging, which mainly adopts averaging or voting strategies to integrate multiple weak learners [70]. The GBDT and XGBoost algorithms are forms of ensemble learning based on boosting, which combines weak learners into a strong learner in an iterative way [71,72].

### 3.2. Shapley Additive Explanations

Shapley additive explanations (SHAP) was proposed by Lundberg and Lee, and is an approach to explain individual and global prediction based on Shapley value [22,73]. The predicted interpretation of an instance ($x$) is estimated by calculating the contribution of each feature to the prediction result of $x$ [22,73]. At present, SHAP is considered to be a recent unified method to explain model prediction, especially "black model" prediction. In

this study, the SHAP interpretable framework was adopted to quantify and attribute the importance of each input feature, and then explore the impact of input feature on prediction of the machine-learning model.

### 3.3. Accumulated Local Effects Plot

In this study, an accumulated local effects (ALE) plot was adopted to visualize and interpret the effect and role of input features (the characteristic value of transverse reinforcement) in the responses of machine-learning models on average. In addition, it was used to identify or estimate the boundary condition between the diagonal splitting failure and shear compression failure. The ALE approach, proposed by Daniel W. Apley and Jingyu Zhu in 2020, can describe how feature averaging affects the prediction of machine-learning models [20,73]. In practice, input features are usually correlated to some extent. The ALE plot is an unbiased and impartial alternative to the partial dependence plot because it overcomes the limitations of the existence of relevant features [20,73].

In theory, the mathematical expression for ALE is defined as follows [20,73]:

$$\begin{aligned} \hat{f}_{x_s, \text{ALE}}(x_s) &= \int_{z_{0,1}}^{x_s} E_{X_c|X_c}\left[\hat{f}^s(X_s, X_c)|X_s = z_s\right]dz_s - constant \\ &= \int_{z_{0,1}}^{x_s} \int_{x_c} \hat{f}^s(z_s, x_c)\mathbb{P}(x_c|z_s)dx_c dz_s - constant \end{aligned}, \tag{3}$$

where $X_s$ is the specified input feature (in this case the characteristic value of transverse reinforcement) and $X_c$ is the set of remaining input features (in this case the shear–span ratio). The ALE plot describes the change in the average prediction, not the prediction itself. The algorithm defines this change as a gradient, but in the actual calculation or operation, replaces it with a predicted difference in an interval [20,73], as follows:

$$\hat{f}^s(x_s, x_c) = \frac{\delta\hat{f}(x_s, x_c)}{\delta x_s}. \tag{4}$$

Based on this, the ALE plot for the characteristic value of transverse reinforcement can be drawn, centered at 0. Therefore, the interpretation observed from the ALE plot is clear and intuitive. Under the condition of a given value, the relative influence of changing the characteristic value of transverse reinforcement on the prediction can also be observed from the ALE plot. In addition to the influencing trend, the ALE plot is an effective approach for identifying the boundary value of the characteristic value of transverse reinforcement [31,33].

Figure 1 also shows the modeling process of this paper. As shown in Figure 1, the experimental dataset was divided into the training set and the testing set. A share of 80% of the data was used for training and 20% of the data were used for testing. Then, the performance evaluation step optimized the key hyperparameters based on a random search and grid search, and compared the prediction accuracy and generalization ability of six machine-learning models. The comparison showed that the ensemble-learning models performed better, and their interpretability was further analyzed. The SHAP approach was used for feature importance and dependency of these models, whereas the ALE plot was used to explore the effect of the change in the characteristic value of transverse reinforcement on model prediction. Moreover, the minimum boundary value of the characteristic value of transverse reinforcement was identified and compared with three mechanism-driven models.

## 4. Prediction of Failure Modes of RC Beam

The work in this section developed prediction models, namely, SVM, DT, MLP, RF, GBDT, and XGBoost, and evaluated their performance. Of the experimental dataset, 80% was used for training machine-learning models, and 20% was used for testing and evaluating model performance. The dataset was randomly divided when the random seed was 0. Furthermore, it is recommended that different models be evaluated and compared

using unified evaluation metrics [69], such as (i) accuracy, (ii) precision, (iii) recall, and (iv) $F_1$-score.

Table 2 lists the results of four evaluation metrics calculated by SVM, DT, MLP, RF, GBDT, and XGBoost. Figure 4 shows model accuracy, that is, the proportion predicted correctly by the model. Generally speaking, the model with a higher accuracy performs better. In this study, for *model* 1 ($\lambda$, $\lambda_{\mathrm{sv},f_t}$) and *model* 2 ($\lambda$, $\lambda_{\mathrm{sv},f_c'}$), the developed GBDT and XGBoost models based on the bagging algorithm have the best performance, followed by the RF models. The GBDT model has an accuracy of 0.86 in the training set and 0.83 in the testing set, and the results of the XGBoost model are the same. Moreover, Figures 5 and 6 show the confusion matrixes on the total dataset obtained by *model* 1 ($\lambda$, $\lambda_{\mathrm{sv},f_t}$) and *model* 2 ($\lambda$, $\lambda_{\mathrm{sv},f_c'}$), respectively, which are used to compare the predicted results with the actual classification results. As shown in the table, columns represent the predicted results and rows represent the actual results. The diagonal elements indicate that the prediction is correct and the non-diagonal elements indicate that the prediction is wrong. The developed DT model based on *model* 1 ($\lambda$, $\lambda_{\mathrm{sv},f_t}$) has the best performance for the category of diagonal splitting failure. These six models have similar performance for the category of shear compression failure. It can also be seen that the GBDT and XGBoost models have the same results on the failure prediction task in this study. Overall, the RF, GBDT, and XGBoost models performed better than the SVM, DT, and MLP models, indicating that ensemble-learning models outperformed traditional models for this dataset. In particular, the GBDT and XGBoost models based on the bagging algorithm have the best prediction accuracy and generalization ability among those investigated in this study.

**Table 2.** The prediction performance of six developed machine-learning models in predicting failure modes of RC beams.

| Feature | Model | Training Set | | | | Testing Set | | | |
|---|---|---|---|---|---|---|---|---|---|
| | | Accuracy | Precision | Recall | $F_1$ | Accuracy | Precision | Recall | $F_1$ |
| $(\lambda, \lambda_{\mathrm{sv},f_t})$ | SVM | 0.71 | 0.67 | 0.71 | 0.66 | 0.69 | 0.64 | 0.69 | 0.65 |
| | DT | 0.86 | 0.86 | 0.86 | 0.86 | 0.80 | 0.79 | 0.80 | 0.79 |
| | MLP | 0.75 | 0.73 | 0.75 | 0.72 | 0.71 | 0.67 | 0.71 | 0.66 |
| | RF | 0.86 | 0.87 | 0.86 | 0.85 | 0.80 | 0.84 | 0.80 | 0.75 |
| | GBDT | 0.86 | 0.87 | 0.86 | 0.85 | 0.83 | 0.86 | 0.83 | 0.80 |
| | XGBoost | 0.86 | 0.87 | 0.86 | 0.85 | 0.83 | 0.86 | 0.83 | 0.80 |
| $(\lambda, \lambda_{\mathrm{sv},f_c'})$ | SVM | 0.70 | 0.49 | 0.70 | 0.58 | 0.71 | 0.51 | 0.71 | 0.59 |
| | DT | 0.86 | 0.86 | 0.86 | 0.86 | 0.78 | 0.77 | 0.78 | 0.77 |
| | MLP | 0.73 | 0.81 | 0.73 | 0.65 | 0.71 | 0.51 | 0.71 | 0.59 |
| | RF | 0.86 | 0.87 | 0.86 | 0.85 | 0.80 | 0.84 | 0.80 | 0.75 |
| | GBDT | 0.86 | 0.87 | 0.86 | 0.85 | 0.83 | 0.86 | 0.83 | 0.80 |
| | XGBoost | 0.86 | 0.87 | 0.86 | 0.85 | 0.83 | 0.86 | 0.83 | 0.80 |

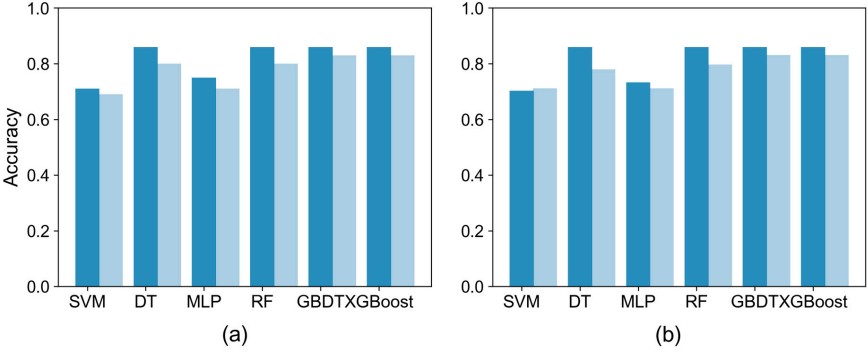

**Figure 4.** The performance (accuracy) for (**a**) *model* 1 ($\lambda$, $\lambda_{\mathrm{sv},f_t}$) and (**b**) *model* 2 ($\lambda$, $\lambda_{\mathrm{sv},f_c'}$).

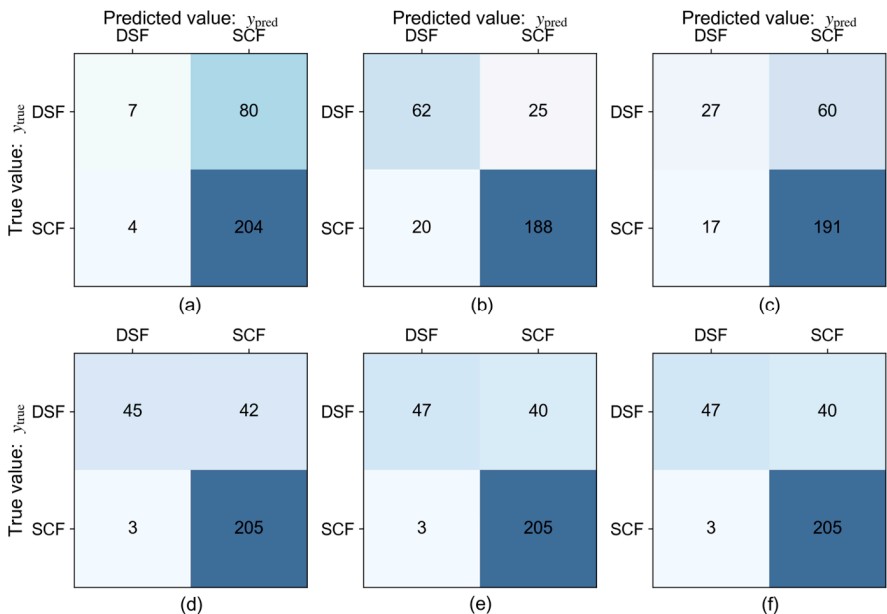

**Figure 5.** Confusion matrixes for *model* 1 ($\lambda$, $\lambda_{sv,f_t}$) based on the total dataset. (**a**) SVM; (**b**) DT; (**c**) MLP; (**d**) RF; (**e**) GBDT; (**f**) XGBoost.

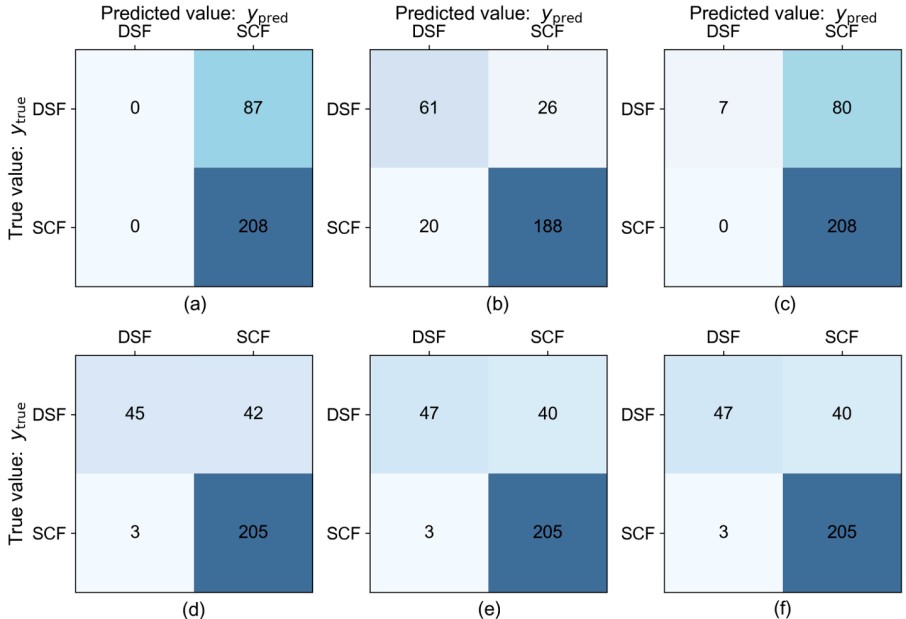

**Figure 6.** Confusion matrixes for *model* 2 ($\lambda$, $\lambda_{sv,f'_c}$) based on the total dataset. (**a**) SVM; (**b**) DT; (**c**) MLP; (**d**) RF; (**e**) GBDT; (**f**) XGBoost.

## 5. Interpretable Analysis

### 5.1. SHAP Feature Importance and Dependency

SHAP can be used to explore the impact of input features on the prediction of RC beams' failure modes by RF, GBDT, and XGBoost. Figure 7 plots the global importance of inputs of *model* 1 ($\lambda$, $\lambda_{sv,f_t}$) and *model* 2 ($\lambda$, $\lambda_{sv,f'_c}$), which is calculated and estimated from the average of the absolute Shapley value for each feature. The higher the average of the absolute SHAP value, the greater the importance of the input feature, that is, the greater the significance of the input feature in model prediction. As seen from Figure 7a, the RF, GBDT, and XGBoost models prefer $\lambda_{sv,f_t}$ to play a more important role in the prediction of failure mode, followed by $\lambda$. Similarly, it is observed that three models prefer $\lambda_{sv,f'_c}$ to play

a more important role in Figure 7b, followed by $\lambda$. The results of previous studies show that $\rho_{\mathrm{stp}}$ has a more significant effect on the failure mode of RC beams [1,4–8].

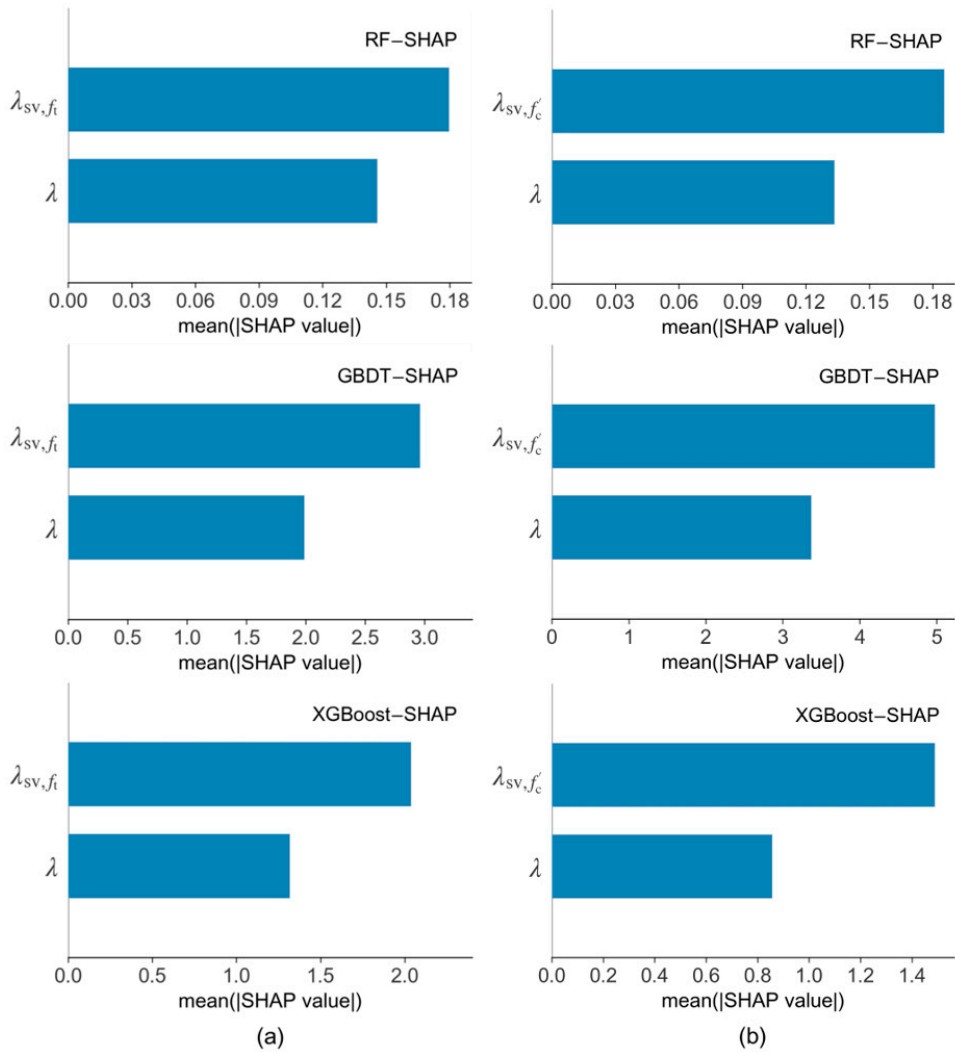

**Figure 7.** SHAP feature importance analysis (Note: The horizontal axis mean ( | SHAP value | ) indicates features' average impact on model output magnitude). (**a**) *model* 1 ($\lambda$, $\lambda_{\mathrm{sv},f_t}$) and (**b**) *model* 2 ($\lambda$, $\lambda_{\mathrm{sv},f_c'}$).

Moreover, the SHAP feature dependency for shear compression failure (SCF) of RC beams is shown in Figure 8, which reflects the interactive influence of $\lambda$ and $\lambda_{\mathrm{sv},f_t}$ (or $\lambda_{\mathrm{sv},f_c'}$) on the model prediction. The $\lambda_{\mathrm{sv},f_t}$ (or $\lambda_{\mathrm{sv},f_c'}$) value is chosen as the horizontal axis, and the SHAP value for $\lambda_{\mathrm{sv},f_t}$ (or $\lambda_{\mathrm{sv},f_c'}$) is chosen as the vertical axis. The $\lambda$ value is shown in color, where a red value indicates a higher value of $\lambda$, while a blue value indicates a lower value of $\lambda$. Based on *model* 1 ($\lambda$, $\lambda_{\mathrm{sv},f_t}$) observed in Figure 8a, the SHAP value for $\lambda_{\mathrm{sv},f_t}$ is negative when $\lambda_{\mathrm{sv},f_t}$ is less than about 0.20, which results in lower probability of SCF of RC beams. When $\lambda_{\mathrm{sv},f_t}$ is greater than 0.20, the SHAP value for $\lambda_{\mathrm{sv},f_t}$ is positive, which contributes to the higher probability of SCF of RC beams. In general, the probability of SCF increases gradually when $\lambda_{\mathrm{sv},f_t}$ changes from 0.00 to 1.20. Based on *model* 2 ($\lambda$, $\lambda_{\mathrm{sv},f_c'}$) observed in Figure 8b, the SHAP value for $\lambda_{\mathrm{sv},f_c'}$ is negative when $\lambda_{\mathrm{sv},f_c'}$ is less than about 0.05, which results in lower probability of SCF of RC beams. When $\lambda_{\mathrm{sv},f_c'}$ is greater than 0.05, the SHAP value for $\lambda_{\mathrm{sv},f_c'}$ is positive, which contributes to the higher probability of SCF of RC beams. In general, the probability of SCF increases gradually when $\lambda_{\mathrm{sv},f_c'}$ changes from 0.00 to 0.70. Although the results obtained by these models are slightly different, the above results

provide an interpretable perspective and insights for the prediction of machine-learning models.

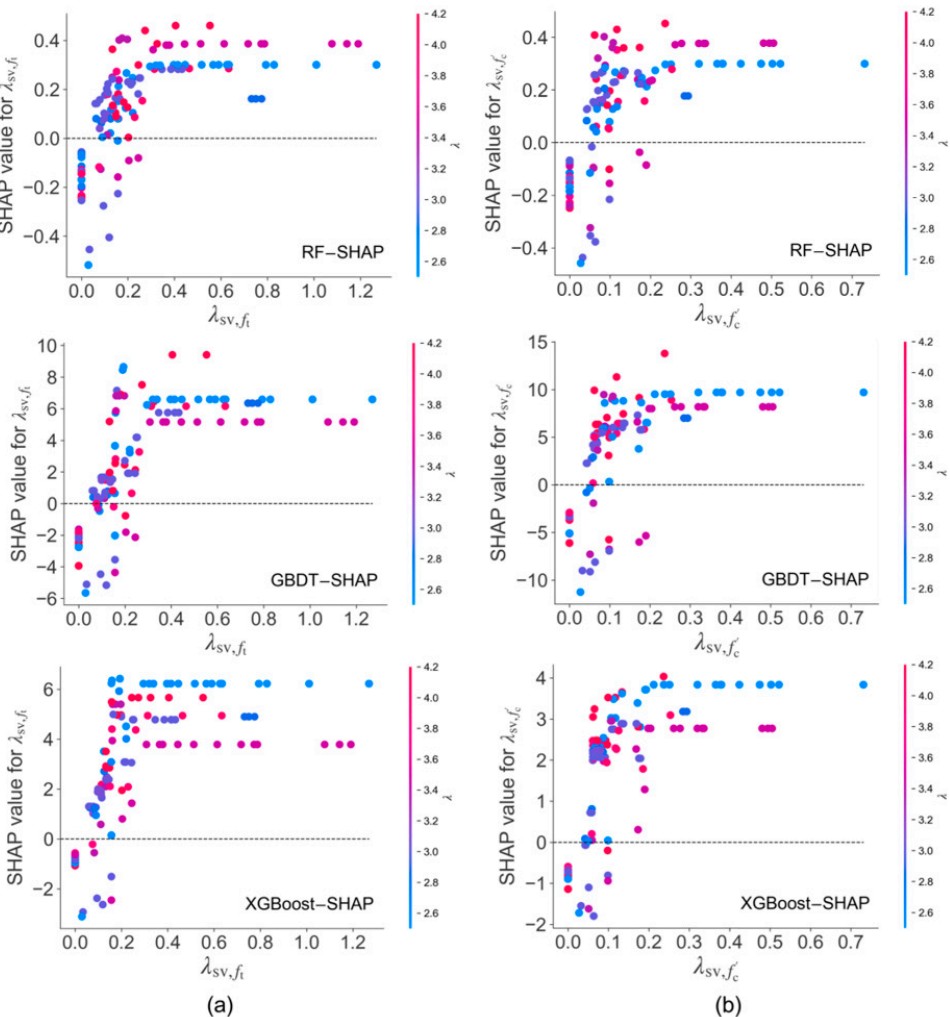

**Figure 8.** SHAP feature dependency for SCF of RC beams. (**a**) *model* 1 ($\lambda$, $\lambda_{sv,f_t}$) and (**b**) *model* 2 ($\lambda$, $\lambda_{sv,f_c'}$).

### 5.2. Identification of the Minimum Characteristic Value of Transverse Reinforcement Based on ALE Plot

For RC beams with transverse reinforcements, the discriminant condition between the diagonal splitting failure and shear compression failure was taken as the characteristic value of transverse reinforcement, which considers the comprehensive effects of the tensile concrete strength, the yield strength for stirrups, and the transverse reinforcement ratio. In addition, the accumulated local effects approach was adopted to explain the effect of the characteristic value of transverse reinforcement on the failure mode and identify the discriminant condition [31,33].

As shown in Figure 9, the effect of $\lambda_{sv,f_t}$ or $\lambda_{sv,f_c'}$ on the failure mode predicted by RF, GBDT, and XGBoost is plotted. The ALE plot is centered at the zero value, which indicates the average model prediction across all variable values. The positive ALE value indicates a higher probability of shear compression failure, while the negative ALE value indicates a lower probability of shear compression failure, which means the probability of diagonal splitting failure is higher. It is observed that the probability of shear compression failure is higher with the larger $\lambda_{sv,f_t}$ in Figure 9a. Based on the RF model, the prediction probability of SCF increases significantly when $\lambda_{sv,f_t}$ is higher than 0.240 according to the sharply rising slope. Moreover, the value of $\lambda_{sv,f_t}$ goes through zero at 0.264, which indicates that 0.264

is the dividing point between DSF and SCF. Similarly, the minimum $\lambda_{\mathrm{sv},f_{\mathrm{t}}}$ value for SCF obtained by GBDT and XGBoost is 0.237 and 0.227, respectively. According to Figure 9b, the minimum $\lambda_{\mathrm{sv},f_{\mathrm{c}}'}$ value for SCF obtained by all of the three ensemble-learning models is 0.059.

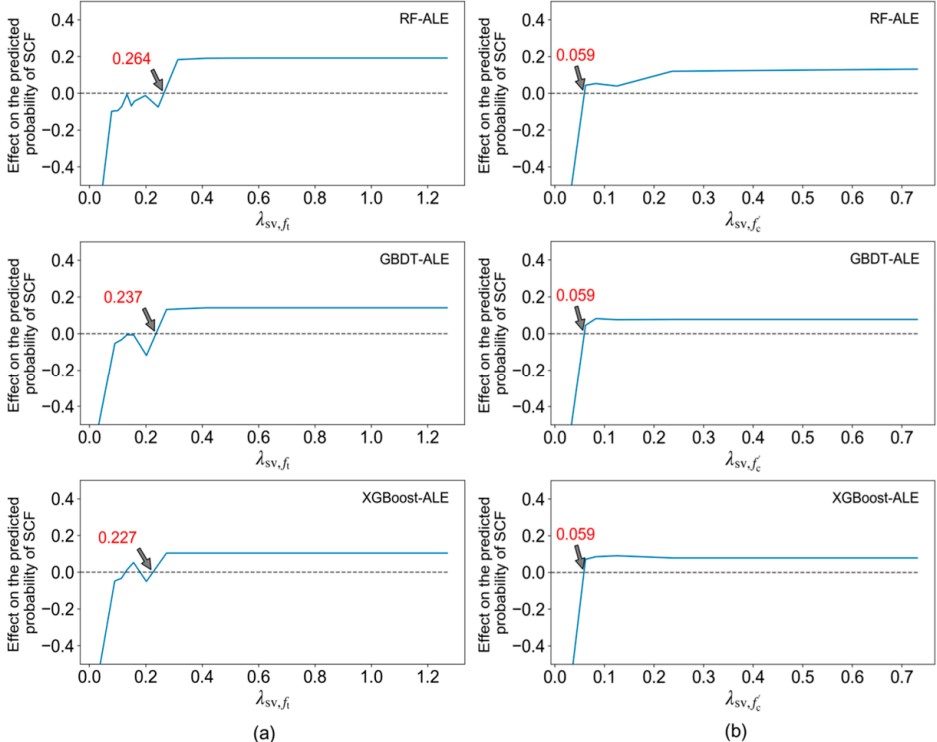

**Figure 9.** ALE plots of $\lambda_{\mathrm{sv},f_{\mathrm{t}}}$ or $\lambda_{\mathrm{sv},f_{\mathrm{c}}'}$ based on RF-ALE, GBDT-ALE, and XGBoost-ALE (Note: The vertical axis indicates centered effect on prediction). (**a**)ALE plots for $\lambda_{\mathrm{sv},f_{\mathrm{t}}}$; (**b**) ALE plots for $\lambda_{\mathrm{sv},f_{\mathrm{c}}'}$.

Some existing results and the results of this paper are listed in Table 3. GB 50010-2010 [34] considers the factor tensile concrete strength ($f_{\mathrm{t}}$), while ACI 318-19 [35] and CSA A23.3-14 [36] consider the factor concrete compression strength ($f_{\mathrm{c}}'$). Overall, the results obtained in this study are comparable with the results of GB 50010-2010 [34], ACI 318-19 [35], and CSA A23.3-14 [36]. Figure 10 also shows that the values of $\lambda_{\mathrm{sv},f_{\mathrm{t}},\mathrm{min}}$ and $\lambda_{\mathrm{sv},f_{\mathrm{c}}',\mathrm{min}}$ obtained in this study are reasonable and the corresponding classification results are superior. Considering $f_{\mathrm{c}}'$, the accuracy of the result ($\lambda_{\mathrm{sv},f_{\mathrm{c}}',\mathrm{min}} = 0.059$) based on data-driven models is even higher, at 0.856. The result has a certain reference and significance.

**Table 3.** The minimum consumption of transverse reinforcement for RC beams with transverse reinforcements and their classification results.

| Type | Approach | The Minimum Consumption of Transverse Reinforcement | Accuracy (Number) |
|---|---|---|---|
| Data-driven | RF-ALE | $\lambda_{\mathrm{sv},f_{\mathrm{t}},\mathrm{min}} = 0.264$<br>$\lambda_{\mathrm{sv},f_{\mathrm{c}}',\mathrm{min}} = 0.059$ | 0.455 (60/132)<br>0.856 (113/132) |
| | GBDT-ALE | $\lambda_{\mathrm{sv},f_{\mathrm{t}},\mathrm{min}} = 0.237$<br>$\lambda_{\mathrm{sv},f_{\mathrm{c}}',\mathrm{min}} = 0.059$ | 0.477 (63/132)<br>0.856 (113/132) |
| | XGBoost-ALE | $\lambda_{\mathrm{sv},f_{\mathrm{t}},\mathrm{min}} = 0.227$<br>$\lambda_{\mathrm{sv},f_{\mathrm{c}}',\mathrm{min}} = 0.059$ | 0.485 (64/132)<br>0.856 (113/132) |
| Mechanism-driven | GB 50010-2010 [34] | $\rho_{\mathrm{stp,min}} = 0.240 f_{\mathrm{t}} / f_{\mathrm{yv}}$ | 0.477 (63/132) |
| | ACI 318-19 [35] | $\rho_{\mathrm{stp,min}} = 0.062 f_{\mathrm{c}}'^{0.5} / f_{\mathrm{yv}}$ | 0.712 (102/132) |
| | CSA A23.3-14 [36] | $\rho_{\mathrm{stp,min}} = 0.066 f_{\mathrm{c}}'^{0.5} / f_{\mathrm{yv}}$ | 0.674 (99/132) |

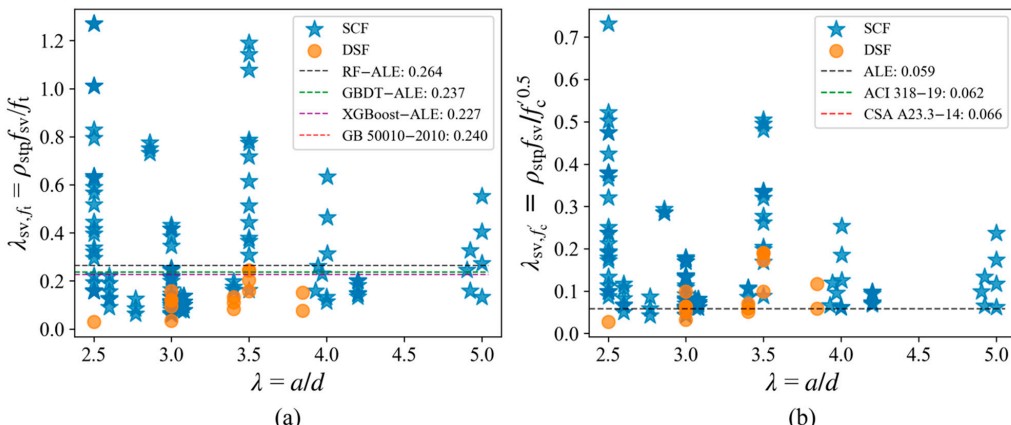

**Figure 10.** The classification results are based on the minimum values of $\lambda_{sv,f_t}$ (**a**) or $\lambda_{sv,f_c'}$ (**b**).

## 6. Conclusions

Based on machine-learning and data-driven approaches, this study attempted to predict failure modes of RC beams and identify the boundary between diagonal splitting failure and shear compression failure. In order to conduct the above research, six machine-learning models were developed to predict failure modes of RC beams. SHAP and ALE methods were utilized to analyze model interpretability and determine the minimum consumption of transverse reinforcement as a boundary between the two failure modes. The main conclusions are as follows:

- To construct prediction models, the shear–span ratio and the characteristic value of transverse reinforcement were selected as model inputs. According to GB 50010-2010, ACI 318-19, and CSA A23.3-14, the characteristic value of transverse reinforcement was constructed in two forms. Given feature subsets, the developed GBDT and XGBoost models had superior performance in predicting failure modes of RC beams, and the prediction accuracy of the training set and testing set was 0.86 and 0.83, respectively.
- According to the shear mechanism, the minimum consumption of transverse reinforcement was designed to avoid diagonal splitting failure. SHAP analysis indicated that the most important feature is the characteristic value of transverse reinforcement, which is reasonable and consistent. Then, ALE was introduced and used to identify the boundary through numerical simulation. Compared with the three codes, the suggested value ($\lambda_{sv,f_c',min} = 0.059$) had the highest security for avoiding diagonal splitting failure. Consequently, this research idea can be extended to similar problems involving parameter threshold identification.

**Author Contributions:** S.W. mainly contributed to conceptualization, methodology, formal analysis, writing—original draft, establishment of database, writing—review and editing, etc. C.M. mainly contributed to the design of this paper, methodology, establishment and analysis of database, formal analysis, writing—original draft, writing—review and editing, supervision, funding acquisition. W.W. mainly contributed to the establishment of the database, formal analysis, resources. X.H. mainly contributed to the establishment of the database. X.X. mainly contributed to methodology, supervision, writing—review and editing. Z.Z., X.L. and J.L. mainly contributed to revision and discussion. All authors have read and agreed to the published version of the manuscript.

**Funding:** This study was financially supported by the NSF of Xinjiang Province (Grant No. 2020Q069, tcbs201928, and 2020D01C066), the Urumqi Outstanding Young Doctor Talent Program. and the Doctoral Foundation of Xinjiang University (Grant No. 620312396).

**Institutional Review Board Statement:** Not applicable.

**Informed Consent Statement:** Not applicable.

**Data Availability Statement:** All necessary data are provided in the article.

**Conflicts of Interest:** The authors declare no conflict of interest.

## Appendix A

**Table A1.** Experimental dataset of RC beams.

| References | Number | Specimens for DSF | Specimens for SCF |
|---|---|---|---|
| [37] | 10 | - | A65-NTR, A65-200, A65-140, A65-95, B65-NTR, B65-200, B65-160, B65-140, B65-125, B65-110 |
| [38] | 3 | S3.0, S4.0 | S2.5 |
| [39] | 12 | R-150, R-300, R-500, R-780, R2-150, R2-300, R2-500, R2-780, R3-150, R3-300, R3-500, R3-780 | - |
| [40] | 11 | S-5.0-A0-1, S-5.0-A0-2, S-5.0-A30-1, S-5.0-A30-2, S-5.0-A60-1, S-5.0-A60-2 | S-2.5-A100, S-3.0-A100-1, S-3.0-A100-2, S-4.0-A100-1, S-4.0-A100-2 |
| [41] | 6 | - | AIC1, AIC2, AIC3, IS1, IS2, IS3 |
| [42] | 10 | - | NC300-52.5, NC300-42.5, SCC300-52.5B1, SCC300-52.5B2, SCC270-52.5B1, SCC270-52.5B2, SCC340-52.5B1, SCC340-52.5B2, SCC380-52.5B1, SCC380-52.5B2 |
| [43] | 7 | - | HA100-I, HA100-II, HA160-I, HA160-II, HA160-III, LA120, LA160 |
| [44] | 4 | - | N-3-1, N-3-2, H-3-1, H-3-2 |
| [45] | 8 | B3-10-1, B3-20-1, B3-30-1, B3-40-1, B3-10-2, B3-20-2, B3-30-2, B3-40-2 | - |
| [46] | 27 | B25-3.0-1, B45-3.0-1, B65-3.0-1, B25-3.4-1, B45-3.4-1, B65-3.4-1, B25-3.0-3, B45-3.0-3, B65-3.0-3, B25-3.4-3, B45-3.4-3, B65-3.4-3 | B25-2.6-1, B45-2.6-1, B65-2.6-1, B25-2.6-2, B45-2.6-2, B65-2.6-2, B25-3.0-2, B45-3.0-2, B65-3.0-2, B25-3.4-2, B45-3.4-2, B65-3.4-2, B25-2.6-3, B45-2.6-3, B65-2.6-3 |
| [47] | 5 | - | R2-1.42-700, R3-1.85-700, R3-1.85-700, R4-1.13-575, R5-1.42-575, R6-1.85-575 |
| [48] | 6 | - | B5N, B6N, B8N, B1ON, B12N, B1OL |
| [49] | 12 | - | M100-S0, M100-S1, M100-S3, M100-S4, M80-S0, M80-S1, M80-S3, M80-S4, M60-S0, M60-S1, M60-S3, M60-S4 |
| [50] | 5 | B11, B12 | B7, B8, B9 |
| [51] | 3 | R-01E, R-02E, R-03E | - |
| [52] | 7 | - | H16S125, H16S155, H16S250, H22S125, H22S155, H22S250, H22S310 |
| [53] | 12 | 2.5-0.00, 3.5-0.00 | 2.5-0.17, 2.5-0.28, 2.5-0.38, 3.5-0.17, 3.5-0.28, 3.5-0.38, 3.5-0.53, 3.5-0.65, 3.5-0.38-D10, 3.5-0.53-D10 |
| [54] | 6 | B3.5-200, B3.5-400, B3.5-700, V-3.5-200, V3.5-400, V3.5-700 | - |
| [55] | 24 | L1-A, L2-A, L3-A, L4-A, L5-A, L6-A, S2-A, S3-A, S4-A, S5-A, S6-A, C1-A | L1-B, L2-B, L3-B, L4-B, L5-B, L6-B, S2-B, S3-B, S4-B, S5-B, S6-B, C1-B |
| [56] | 3 | - | Case-1, Case-2, Case-3 |
| [57] | 12 | - | A1, A2, B1, B2, C1, C2, TA1, TA2, TB1, TB2, TC1, TC2 |
| [58] | 7 | - | S1-25-05, S2-25-25, S3-25-50, S4-25-75, S2-40-25, S3-40-50, S4-40-75 |
| [59] | 5 | WB-1, WB-2, WB-3, WB-4, WB-5 | - |

**Table A1.** *Cont.*

| References | Number | Specimens for DSF | Specimens for SCF |
|---|---|---|---|
| [60] | 15 | - | NA-S2, NA-M2, NA-L2, NA-M3, NA-L4, RH-S2, RH-M2, RH-L2, RH-M3, RH-L4, RF-S2, RF-M2, RF-L2, RF-M3, RF-L4 |
| [61] | 18 | - | CC NS-4(1), CC NS-4(2), CC NS-6(1), CC NS-6(2), CC NS-8(1), CC NS-8(2), RAC50 NS-4(1), RAC50 NS-4(2), RAC50 NS-6(1), RAC50 NS-6(2), RAC50 NS-8(1), RAC50 NS-8(2), RAC100 NS-4(1), RAC100 NS-4(2), RAC100 NS-6(1), RAC100 NS-6(2), RAC100 NS-8(1), RAC100 NS-8(2) |
| [62] | 9 | - | NAC-1, RAC50-1, RAC100-1, NAC-2, RAC50-2, RAC100-2, NAC-3, RAC50-3, RAC100-3 |
| [63] | 10 | 1, 8 | 2, 3, 4, 5, 6, 7, 9, 10 |
| [64] | 18 | H50/1, H50/5, H60/1, H75/1, H100/1, H100/5 | H50/2, H50/3, H50/4, H60/2, H60/3, H60/4, H75/2, H75/3, H75/4, H100/2, H100/3, H100/4 |
| [65] | 3 | - | B60-2, B60-8, B30-2 |
| [66] | 8 | S5, S3, S2, OI-2, S3k, S5k, S1, OI-1 | - |
| [67] | 9 | MHB2.5-0 | MHB2.5-25, MHB2.5-50, MHB2.5-75, MHB2.5-100, HB2.5-25, HB2.5-50, HB2.5-75, HB2.5-100 |

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
