# Peer review of "Prediction of Failure Modes and Minimum Characteristic Value of Transverse Reinforcement of RC Beams Based on Interpretable Machine Learning"

_buildings, doi:10.3390/buildings13020469_

Round 1

Reviewer 1 Report

In this manuscript, the interpretable machine learning approach is employed to determine the possible failure modes of RC beams. Design suggestions to avoid diagonal splitting failure are proposed according to the analysis results. Therefore, this research work is interesting and significant. However, the manuscript has a few shortcomings which need to be addressed before its publication. The following comments may help the authors to improve the quality of this paper.

1.       Abstract and Introduction: For international readers, it may well be difficult to understand the GB50010-2010”. It is thus suggested to use “China Design Code of GB50010-2010” instead.  Similarly, it is suggested to use “Canada Design Code of CSA A23.3-14” instead of “CSA A23.3-14”.

2.       There are a few writing problems, such as “On the one hand”, “reinforcements”, “shapley additive explanations” in Line 343, and so on. Please polish the manuscript to further improve the attraction of this paper.

3.       Test results from a total of 295 RC beams were applied to construct the experimental dataset. Hence the dataset is convincing. As shown in Figure 1, the experimental dataset contains two parts, namely slender RC beams with and without transverse steel reinforcement. In future studies, the authors are suggested to focus on all the RC beams instead of slender ones only.

4.       It is uncommon to use the square root of MPa, as can be seen in Line 171 and Table 1. Please add some explainations to such expressions.

Author Response

  The authors thank the reviewer for the valuable and careful comments. All of the following comments have been addressed in the revised manuscript.

  In order to improve the quality of this paper, the full test has been checked and corrected in the revised manuscript (e.g. Lines 23-25, 60, 86-87, 179, 190, 192, 221, 224 et al.).

Reviewer 2 Report

The reviewed paper, "Prediction of failure modes and minimum characteristic value of transverse reinforcement of RC beams based on interpretable machine learning," is interesting and informative. This paper uses machine learning techniques to predict failure modes and identify boundary values to avoid diagonal splitting failures. This is a well-written paper. Figures, tables, and analysis results are well presented. This paper provides a good discussion and clear conclusions.

The following are some suggested points to make the paper easier to understand:

1. In Section 1, it is advisable to add a drawing showing the difference between diagonal splitting failure (DSF), shear compression failure, and diagonal compression failure (SCF).

2. More detailed information for the prediction model in Section 4 is needed to clarify further the results obtained in Figure 6.

3. The flowchart or algorithm of one of the machine learning used in this paper should be explained in the text.

Author Response

  The authors thank the reviewer for the valuable and careful comments. All of the following comments have been addressed in the revised version of the manuscript.

Reviewer 3 Report

1- The abstract is not clear, please rewrite it.

2- This paper talks about shear mood failure so this should be clear in the main title of the paper.

3- In line 124, show samples for DSF and SCF.

4- In line 171 I don't think it is proper to use 0.0 square root of MPa, change it. just use 0.0.

5- Conclusions are unclear and very general. It is a must that you provide a better quality presentation for them.

Author Response

(The authors gave the same response as above.)
